# Expression of Neurokinin B Receptor in the Gingival Squamous Cell Carcinoma Bone Microenvironment

**DOI:** 10.3390/diagnostics11061044

**Published:** 2021-06-07

**Authors:** Shoko Yoshida, Tsuyoshi Shimo, Kiyofumi Takabatake, Yurika Murase, Kyoichi Obata, Tatsuo Okui, Yuki Kunisada, Soichiro Ibaragi, Hitoshi Nagatsuka, Akira Sasaki

**Affiliations:** 1Department of Oral and Maxillofacial Surgery, Okayama University Graduate School of Medicine, Dentistry and Pharmaceutical Sciences, Okayama 700-8525, Japan; de17050@s.okadai.jp (Y.M.); de18016@s.okayama-u.ac.jp (K.O.); kunisada.y@s.okayama-u.ac.jp (Y.K.); sibaragi@md.okayama-u.ac.jp (S.I.); aksasaki@md.okayama-u.ac.jp (A.S.); 2Division of Reconstructive Surgery for Oral and Maxillofacial Region, Department of Human Biology and Pathophysiology, School of Dentistry, Health Sciences University of Hokkaido, Hokkaido 061-0293, Japan; 3Department of Oral Pathology and Medicine, Okayama University Graduate School of Medicine, Dentistry and Pharmaceutical Sciences, Okayama 700-8525, Japan; gmd422094@s.okayama-u.ac.jp (K.T.); jin@okayama-u.ac.jp (H.N.); 4Department of Oral and Maxillofacial Surgery, Faculty of Medicine, Shimane University, Izumo, Shimane 693-8501, Japan; tatsuookui0921@gmail.com

**Keywords:** neurokinin B receptor, gingival squamous cell carcinoma, osteoclast

## Abstract

Gingival squamous cell carcinoma (SCC) frequently invades the maxillary or mandibular bone, and bone destruction is known as a key prognostic factor in gingival SCCs. Recently, Neurokinin 3 receptor (NK-3R), the receptor ligand for NK-3, which is a member of the tachykinin family expressed in the central nervous system, was identified through pathway analysis as a molecule expressed in osteoclasts induced by the hedgehog signal. Although the expression of NK-3R has been detected in osteoclast and SCC cells at the bone invasion front, the relationship between NK-3R expression and the prognosis of gingival SCC patients remains unclear. In the present study, we retrospectively reviewed 27 patients with gingival SCC who had undergone surgery with curative intent. Significantly higher NK-3R expression in tumor cells was found in a case of jawbone invasion than in a case of exophytic poor jawbone invasion. On the other hand, no significant association was observed between NK-3R tumor-positive cases and tumor size, TNM stage, or tumor differentiation. The survival rate tended to be lower in NK-3R tumor-positive cases, but not significantly. However, the disease-specific survival rate was significantly lower in patients with a large number of NK-3R-positive osteoclasts than in those with a small number of them at the tumor bone invasion front. Our results suggest that NK-3R signaling in the gingival SCC bone microenvironment plays an important role in tumor bone destruction and should be considered a potential therapeutic target in advanced gingival SCC with bone destruction.

## 1. Introduction

Oral neoplasms account for 5% of all human cancers, and oral squamous cell carcinoma (OSCC) is the most common malignant tumor of the oral cavity. The rate is less than 0.1% in some countries and over 40% in others, and it is ranked as the sixth most common tumor in the world [1]. Despite advances in the diagnosis and treatment of OSCC, patients still have poor survival and quality of life, often with loss of speech, difficulty chewing or swallowing, pain, facial deformity, and severe psychological problems [2]. The gingiva is the third most common site of OSCC, after the tongue and floor of the mouth [3]. However, gingival squamous cell carcinoma (SCC) poses a diagnostic challenge to clinicians because the inspection for it is very similar to that for periodontitis and is often diagnosed as advanced cancer. Gingival SCC arises from the mucosal surface of the oral gingiva and frequently invades the maxillary or mandibular bone [4,5,6]. Recently, bone invasion was found to be a key independent prognostic factor in gingival SCC [4,7]. In gingival SCC bone destruction, paracrine Hedgehog signaling in tumor cells is involved in indirect osteoclastogenesis, which promotes osteoclast formation by upregulating RANKL production in bone marrow stromal cells [8]. On the other hand, osteoclasts at the cancer invasion front express patched1 and Gli-2, and SHH promotes osteoclast differentiation and bone resorption through direct action on osteoclast precursor cells [9]. To identify the molecular changes that occur when osteoclasts are initiated by Hedgehog signaling, a microarray analysis was performed, and pathway analysis revealed that the Neurokinin 3 receptor (NK-3R) in non-odorant G-protein-coupled receptors (GPCRs) was the most upregulated gene stimulated by Hedgehog signaling [9]. NK-3R is expressed mainly in the central nervous system and is the most selective of the tachykinin receptors, with highly preferential binding and activation by its endogenous ligand neurokinin B, NKB (tachykinin 3, *TAC3*) [10,11]. NKB is a member of the tachykinin family of peptides, which are characterized by a common carboxyl-terminal region, Phe-X-Gly-Leu-Met-NH_2_, and which includes substance P, neurokinin A, and NKB [12]. The NKB/NK-3R system is expressed in mammals, and its expression and function vary with age, menstrual cycle, and the reproductive biology field [13,14]. In pathological conditions, NK-3R was expressed in human OSCCs in head and neck tumors and had a function of bone destruction [15,16]. It is still unclear, however, how NK-3R signaling participates in osteoclastic bone resorption by OSCC cells in the clinic. The purposes of this study were to clarify the expression of NK-3R in gingival SCC using clinical resection specimens and to determine whether NK-3R is a prognostic factor in gingival SCC.

## 2. Patients and Methods

### 2.1. Patients

The 27 patients included in the study had been diagnosed and treated for upper or lower gingival SCC individually at the Okayama University Hospital (Okayama, Japan), Department of Oral and Maxillofacial Surgery (Biopathology) from 2011 through to 2015. All patients were followed until death or the end of 2018. The surgically resected tumors were collected as part of routine care by the authors. No patient had received chemotherapy or radiation therapy before surgery. Table 1 summarizes the details of the gingival SCC patient data, including age, gender, TNM staging, differentiation, the incidence of recurrence, and secondary lymph nodal metastasis. The follow-up periods ranged from 5.8 to 83.4 months (mean 52.1 months). During the follow-up, five patients with local cancer recurrence and three patients with lymph node/distant metastasis were observed. Of the 27 patients, three succumbed to cancer. The tissue samples were fixed with 10% neutral-buffered formalin and processed for paraffin-embedded sections. Serial sections (4 μm) were used for hematoxylin and eosin (H & E) sections and immunohistochemical analysis.

Each tumor was clinically staged according to the TNM Classification of Malignant Tumors (8th edition), defined by the UICC [17]. The grade of tumor differentiation was based on the WHO criteria for histological typing of oral and oropharyngeal tumors [1]. This retrospective study was approved by the Ethics Committee of the Okayama University Graduate School of Medicine, Dentistry, and Pharmaceutical Sciences (Protocol No: 1612-012, 14 December 2016).

### 2.2. Histopathological Analysis

H & E staining was performed on the resected specimens, and the boundary between the tumor and the bone was observed. Bone invasion was categorized as no bone invasion when there was a fibrous connective tissue layer at the boundary between the tumor and bone, cortical when the bone invasion was limited to the cortex, and medullary when extension into the cancellous bone was present. In this study, the cortical invasion was not classified as T4a because the tumor does not invade through the cortical bone.

### 2.3. Immunohistochemical Analysis

The IHC analysis was performed with anti-NK-3R antibody at a 1:100 dilution (#bs-0166R, anti-rabbit IgG, Bioss, Woburn, MA, USA). Paraffin blocks of specimens were cut at 5 μm thickness. Briefly, cut tissue sections were deparaffinized in xylene for 20 min and then dehydrated in graded alcohol solutions. The endogenous peroxidase activity was then blocked by incubation in 1.25% H_2_O_2_ in methanol for 30 min. For antigen retrieval, the sections were heated in 0.01 mol/L citrate buffer (pH 9.0) for 15 min. After reaching room temperature, the sections were incubated with the primary antibodies at 4 °C for 16 h and visualized with the Envision system (Dako, Tokyo, Japan) for 1 h. Sections then reacted with 0.02% 3,3′-diaminobenzidine (Wako, Osaka, Japan) with 0.006% H_2_O_2_ in 0.05 mL/L Tris-HCl. The sections were counterstained by Mayer’s hematoxylin and mounted.

### 2.4. Double-Fluorescent Immunohistochemical Analysis

Double-fluorescent IHC for NK-3R-CD68 was performed using anti-NK-3R antibody (1:100) (#bs-0166R, anti-rabbit IgG, Bioss) and anti-CD68 (1:100) (#KP1, anti-mouse IgG, Dako, Glostrup, Denmark). The secondary antibodies applied were anti-rabbit IgG Alexa Flour 568 and anti-mouse IgG Alexa Flour 488 (Thermo Fisher, Tokyo, Japan). Antibodies were diluted with Can Get Signal A (TOYOBO, Osaka, Japan). After antigen retrieval, sections were treated with Block Ace (DS Pharma Bio-medical, Osaka, Japan) for 20 min at room temperature. Specimens were incubated with primary antibodies at 4 °C overnight and then incubated with secondary antibodies (1:200) for 1 h at room temperature. After the reaction, the specimens were stained with 1 mg/mL DAPI (Dojindo Laboratories, Kumamoto, Japan).

### 2.5. Quantification of NK-3R in Tissue Sections

The degree of primary antibody reactivity on individual tissue sections was scored semi-quantitatively (percentage of stained tumor cells and osteoclasts in the sections) by three observers, as described previously [18]. The population of immunoreactive tumor cells at the invasion front at the bone destruction site was subdivided as follows: the immunoreactivity of the anti-NK-3R antibody was graded according to the number of stained cells, and the staining intensity in individual cells was as follows: <50% weak, ≥50% weak, and <50% strong were classified as negative, and >50% strong was classified as positive. The NK-3R-positive osteoclasts were counted by taking a ×400 randomly photographed tumor bone invasion front as one field of view, counting the NK-3R and CD68 double-positive osteoclasts contained in it, and averaging the three fields of view. The presence of three or more NK-3R-positive osteoclasts in one visual field was defined as a large number of cases, and that of two or fewer was defined as a small number of cases. All the multiple serial sections in a clinical specimen were checked by surgeons and pathologists, confirming that the double-positive analyzed data were consistent with HE data. Figure 1 and Figure 2 show the results of representative NK-3R and CD68 staining in the tumor bone invasion front.

### 2.6. Statistical Analysis

Cross-tabulated data were analyzed using the chi-square test or Fisher exact test, where appropriate. The overall survival (OS) and disease-specific survival (DSS) rates were calculated by Kaplan–Meier methods, and statistical significance was analyzed by the log-rank test. Values of p < 0.05 were considered statistically significant.

## 3. Results

### 3.1. Patient Demographics

Table 1 shows the patients’ clinicopathological characteristics. The 27 cases of gingival SCC included 10 males (37.0%) and 17 females (63.0%). The affected areas were the upper gingiva (10 cases, 37.0%) and the lower gingiva (17 cases, 63.0%). The patients’ average age was 72.8 years (range of 39–89 years), and the median follow-up period was 52.1 months. Tumor characteristics were as follows. Size: <2 cm (14.8%), ≥2 cm and <4 cm (63.0%), and ≥4 cm (22.2%). For lymphatic metastasis (pathological N stage): N0 (81.5%), N1 (11.1%), and N2 (7.4%). For the TNM stage: stage I (11.1%), stage II (48.1%), stage III (3.7%), stage IVA (33.3%), and stage IVB (3.7%). For tumor differentiation: well (55.6%), moderate (33.3%), and poor (11.1%).

### 3.2. NK-3R Expression in Tumor Cells and Clinicopathological Characteristics in Human Gingival SCC

First, we investigated the relationships between NK-3R expression in tumor cells and various clinical features of gingival SCC. Based on our statistical analyses, we assessed the correlations between NK-3R expression and clinicopathologic parameters, including age, gender, primary tumor size, lymphatic metastasis, TNM stage, tumor differentiation, bone invasion pattern, lymphovascular invasion, perineural invasion, primary recurrence, and secondary cervical lymph nodal metastasis (Table 2). Among the 27 patients, positive NK-3R expression was not significantly associated with age (*p* = 0.41), gender (*p* = 0.13), primary tumor size (*p* = 0.08), lymphatic metastasis (*p* = 0.59), TNM stage (*p* = 0.49), tumor cell differentiation (*p* = 0.08), lymphovascular invasion (*p* = 0.37), perineural invasion (*p* = 0.44), primary recurrence (*p* = 0.96), or secondary cervical lymph nodal metastasis (*p* = 0.61). However, NK-3R expression was significantly associated with thebone invasion pattern, and there was a significant difference in NK-3R expression in tumor cells in the cases with compressive patterns of cortical bone absorption and in those with medullary bone destruction (*p* = 0.03).

### 3.3. NK-3R Expression in Osteoclasts and Clinicopathological Characteristics in Human Gingival SCC

Next, we investigated the relationship between the number of NK-3R-positive osteoclasts and various clinical features of gingival SCC. Based on statistical analyses, we assessed the correlation between the number of NK-3R-positive osteoclasts at the tumor invasion front and the clinicopathologic parameters (Table 3). Among the 25 patients, nine had large numbers of NK-3R-positive osteoclasts, and 16 had small numbers of them (Table 3). The presence of a large number of NK-3R-positive osteoclasts was not significantly associated with age (*p* = 0.28), gender (*p* = 0.09), primary tumor size (*p* = 0.49), lymphatic metastasis (*p* = 0.12), tumor cell differentiation (*p* = 0.97), lymphovascular invasion (*p* = 0.07), perineural invasion (*p* = 0.23), primary recurrence (*p* = 0.71), or secondary cervical lymph node metastasis (*p* = 0.91). However, the presence of a large number of NK-3R-positive osteoclasts was significantly associated with TNM stage (*p* = 0.016) and the medullary bone invasion pattern (*p* = <0.001). There was no significant association with the presence of NK-3R-positive tumor cells at the invasion front (*p* = 0.10, Table 3).

### 3.4. NK-3R Expression in Osteolytic Gingival SCC and Osteoclasts

Figure 1 shows a microscopic image of the NK-3R expression pattern representative of invasive bone destruction observed in patients with OSCC in the mandibular region. NK-3R in gingival cancer invading the jawbone was divided into an NK-3R weak expression group (Figure 1A–C) and an NK-3R strong expression group (Figure 1D–F). Each photo on the right shows a magnification of the rectangle-delimited area of cancer bone invasion in the corresponding left photo. In the NK-3R strong expression group (Figure 1D–F), it could be seen that NK-3R expression in the cytoplasm of osteoclasts at the bone destruction site was also higher than that in the NK-3R weak expression group (Figure 1A–C). Excluding cases in which there were low numbers of stromal cells between the tumor and bone, and cases in which the tumor cells nested directly into the resorbing bone, it was found that high expression of NK-3R in the invasive front of cancer cells at the bone destruction site resulted in high expression of NK-3R in osteoclasts (10 out of 16 cases). On the other hand, most of the osteoclasts with low NK-3R expression were those with no bone invasion or resorption to cortical bone and involved cases where the tumor proliferated outward or was poorly invasive. Both the NK-3R weak expression group (Figure 1A–C) and the NK-3R strong expression group at the bone invasion front (Figure 1D–F) showed weak NK-3R expression on the epithelial surface, which is the oral side of the tumor in mandibular gingival cancer. This is consistent with our previous report (data not shown) [15].

To determine whether the NK-3R is accurately and specifically expressed in the osteoclasts at the invasive bone destruction site observed in patients with gingivalSCCs, we performed double-fluorescent IHC for NK-3R-CD68 (Figure 2A–D). As shown in Figure 2, the expression of NK-3R was observed not only in the tumor cells (Figure 2A Tm and Figure 2B), but also in the almost osteoclasts (Figure 2A OC and Figure 2B), which were correlated with the expression of CD68-positive cells (Figure 2C,E, arrowheads) at the tumor bone invasion front.

### 3.5. Survival Analysis

First, to determine whether the status of NK-3R expression had an effect on prognosis, we evaluated the significance of NK-3R expression in tumor cells using the Kaplan–Meier method, and the difference in follow-up times between the two groups (that with positive and that with negative expression of NK-3R) was compared using the log-rank test. Patients with positive NK-3R tumors in overall survival and disease-specific survival had no correlation with patients with negative ones: *p* = 0.36 and *p* = 0.46, respectively (Figure 3A,B). Next, to determine the effect of the number of NK-3R-positive osteoclasts at the tumor invasion front on prognosis, we evaluated the significance of NK-3R expression in tumor cells using the Kaplan–Meier analysis. The patients with a large number of positive NK-3R osteoclasts tended to show lower overall survival than the patients with a small number of them, *p* = 0.22 (Figure 4A). On the other hand, the patients with a large number of positive NK-3R osteoclasts showed significantly lower disease-specific survival, *p* = 0.05 (Figure 4B).

## 4. Discussion

NK-3R, a tachykinin receptor, is a GPCR that is preferentially activated by Hedgehog agonists in osteoclasts [9]. In addition to being involved in a wide range of CNSs, it can exert many biological effects, including smooth muscle contraction and relaxation, vasodilation, immune system activation, pain transmission, and neurogenic inflammation [19]. In this study, we defined variables that affect local recurrence and survival and determined whether the number of NK-3R-positive cancer cells and osteoclasts is a prognostic factor for gingival SCC. We focused on three types of jawbone infiltration patterns in conducting these studies. Our previous research found that there was no significant association between tumor location (maxillary vs. mandibular) and bony invasion, despite well-known differences in bone density [7]. In the present study, the maxilla and mandible were examined without being separated. In gingival cancer cells, using mandibular resection specimens, it was found that NK-3R was significantly expressed in the cortical bone invasion pattern and medullary bone invasion pattern compared with the no bone invasion type (Table 2). This suggests that the expression of NK-3R in cancer cells is affected not only by the individual characteristics of the cancer cells themselves, but also by the cancer microenvironment. In the cortical bone invasion pattern, some factors are produced from the bone by osteoclasts appearing on the outer surface of cortical bone. In the medullary bone invasion pattern, on the other hand, tumor cells are more aggressive than those in the cortical bone invasion pattern. The tumor invades the bone marrow through the enlarged Volkmann and Haversian tubes, via the periodontal cavity, or by the direct resorption of cortical bone. From these findings, we can infer the existence of factors supplied by the bone marrow environment.

NK-3R expression was observed in tumor cells that had invaded the bone matrix but not in tumor cells on the oral side OSCC (*p* < 0.01). These results are consistent with previous results (data not shown) [15]. Since the expression of SHH in gingival SCC cells using mandibular resection specimens was also increased in the tumor invasion front [9], it is suggested that the NK-3R signal is involved in the infiltration and proliferation of cancer cells enhanced by the autocrine action of SHH produced from cancer cells in the tumor invasion front in the bone marrow environment. Unlike the cases of no invasion pattern or cortical bone invasion pattern, the medullary bone invasion pattern is an independent predictor of reduced overall survival and disease-specific survival [7], suggesting that NK-3R expression in cancer cells is associated with the prognosis of gingival SCC. However, regarding the examination of the relationship between the NK-3R expression and prognosis of gingival SCC patients, it is necessary to further increase the number of cases and examine it in detail in the future.

TNM stage is the factor that strongly influences the prognosis of OSCC patients [20]. Even when tumor size, nodal disease, and metastases were controlled for, T4 disease was an independent predictor of survival and portended a worse disease-specific survival [2]. However, there was no correlation between NK-3R expression in cancer cells and the TNM stage, and no association was found between NK-3R expression in cancer cells and prognosis in either overall survival or disease-specific survival. On the other hand, among the clinical parameters, the numbers of NK-3R-positive osteoclasts differed significantly between TNM stages III and IV compared with stages I and II (Table 3, *p* = 0.016), and it could be seen that the number of NK-3R-positive osteoclasts in the medullary bone invasion pattern was significantly higher than that of the combined group of no bone invasion pattern and cortical bone invasion pattern (Table 3, *p* < 0.001). Elmusrati et al. reported that no direct contact between tumor and bone was evident, with abundant fibrous and variably desmoplastic stroma present at the OSCC tumor-bone interface in the majority of the cases in the cohort they analyzed [21]. The existence of cancer-associated fibroblasts at the tumor bone invasion front is associated with more destructive behavior, and this is consistent with our observation that the presence of large numbers of osteoclast and stromal cells at the surface of the tumor bone invasion front suggested higher levels of bone turnover, bone resorption, and aggressiveness of OSCC. Interestingly, two out of 27 cases with a medullary bone invasion pattern showed the direct bone destruction of tumor cells and did not show the presence of stromal cells and osteoclasts. Those two cases were excluded from the analysis of the number of NK-3R-positive osteoclasts shown in Table 3. The number of tartrate-resistant acid phosphatase (TRAP)-positive osteoclasts increased during the transition from non-proliferative to overt osteolytic metastases, whereas the percentage of osteoclastic cells surrounding breast cancer micro-metastasis was below 20% [22]. Cathepsin-K-positive osteoclasts were not found to be in direct contact with the cancer cells [23]. In addition, the inhibition of osteoclasts by zoledronic acid did not inhibit or reduce breast cancer cell homing to bone in vivo [24], suggesting that osteoclasts probably do not regulate tumor cell homing to bone [22]. These results suggest that it would be necessary to clarify the mechanism underlying the direct interaction between OSCC tumor cells and bone tissue, including released factors in clinical and basic research. It has also been suggested that the roles of NK-3R in tumor cell invasion and proliferation ability, as well as osteoclast bone resorption, act by different spatiotemporal mechanisms, because no significant correlation was found between the expression of NK-3R in cancer cells and the number of NK-3R-positive osteoclasts (Table 3, *p* = 0.01). The functional mechanism underlying the NKB signal in the tumor remains to be examined in the future.

Osteoclasts play an important role in bone destruction in gingival SCC. To clarify the role of NK-3R expression upregulation in osteoclasts by the SHH ligand, it is important to resolve the role of the SHH signal in osteoclasts. Activation of Hh signaling in mature osteoblasts by the conditional deletion of Ptch1 induces RANKL expression in osteoblasts and indirectly stimulates osteoclast formation [25,26,27]. In the tumor bone microenvironment, the Hh ligand produced by tumor cells induces RANKL production in osteoblasts and indirectly promotes osteoclast formation [8,28,29]. Together, these reports suggest that osteoclast formation promoted by a non-cell-autonomous mechanism is mediated by osteoblasts in the tumor-induced bone destruction microenvironment. However, the cell-autonomous role of Hh signaling in osteoclast formation remains unclear. Li et al. showed that the stimulatory effect of Hedgehog signaling on osteoclast formation and the inhibitory effect on osteoclast apoptosis were dependent on the Gli family of transcription factors [30]. Similarly, we demonstrated that Shh enhanced RANKL-induced osteoclast differentiation and that the effects of SHH on RANKL-induced osteoclastogenesis were almost completely abolished by the ablation of Smoothened [9]. On the other hand, conditional ablation of Ihh in limb mesenchymal cells, which potentially disrupted Hh signaling in both osteoblastic and osteoclastic lineage cells, led to increased osteoclast formation [31]. Recently, Zang et al. demonstrated that Sufu deletion or treatment with Shh signaling agonist, purmorphamine, not only potently inhibited osteoclast differentiation in vitro, but also strongly suppressed titanium particle-induced osteoclastogenesis in vivo [32]. These reports suggest that Hh signaling may play the opposite role in osteoclast formation under different conditions. Hh pathway activation is determined by the balance between bone formation and bone resorption and is thought to function differently depending on the physiological or disease-specific pathological condition. It is suggested that the Hh signal in the microenvironment in OSCC jawbone destruction is complicated and that the destruction may be regulated through a non-cell-autonomous mechanism, rather than through a cell-autonomous role, suggesting the importance of the presence of NKB signals. The mechanism involved in osteoclastic bone resorption via NK-R3 is still unknown. NKB was localized in peripheral neurons and may involve the activation of osteoclast formation and bone resorption through NK-3R [33]. SB222200, a selective antagonist of NK-3R, significantly suppressed the radiographic osteolytic lesion and tumorigenesis in the tibial metaphysis bearing human OSCC HSC-2 cells in mouse models. It was suggested that NK-3R signaling is a potential target for the treatment of OSCCs in cases of bone destruction [16]. As shown in Figure 2, the expression of NK-3R was observed not only in the tumor cells, but also in the almost osteoclasts, which were correlated with the expression of CD68-positive cells at the tumor bone invasion front. However, regarding whether the prognosis of patients could be predicted simply by counting the number of osteoclasts by H & E staining, regardless of the expression of NK-3R, further detailed study is needed, because these data included both weak and strong expression of NK-3R in osteoclasts.

Increased serum levels of NKB were observed in a patient with premature thelarche and central precocious puberty [34] and in pregnant females [35]. NKB is thought to be produced from blood vessels in the tumor bone microenvironment. Positive lymphovascular invasion indicated poor survival and lymph nodal metastasis trends, and lymphovascular invasion might be used as a prognostic biomarker for patients with OSCC, in addition to the TNM staging system [36]. Pal et al. demonstrated that NKB could target the endothelium via a multi-component mechanism to oppose vascular remodeling [37], and Wang et al. reported that NK-3R agonist analogues might be used as templates for the development of anti-tumor drugs [38].

On the other hand, as the blood supply of bone is accompanied by innervation, it has been demonstrated that the peripheral nervous system is involved in bone metabolism, osteogenic differentiation, bone mineralization, and bone remodeling [39]. Perineural invasion can be found in a variety of malignant tumors and may be a sign of tumor metastasis and invasion; it portends a poor prognosis. The pathology and clinical significance of perineural invasion are clearly understood, but exploration of the underlying molecular mechanism is ongoing [40]. NKB provides an ideal nodal point through which peripheral signals can alter gonadotropin secretion and reproductive capacity [41]. NK-1R ligand substance P is widely expressed in the peripheral nervous system, and it plays important roles in the development of pancreatic cancer metastasis and perineural invasion [42]. Perineural invasion was identified in 17.4% of OSCCs, and it is significantly associated with morbidity and mortality, and it confers a poor prognosis [43,44,45]. The interaction network between the tumor and the NKB secreted by the nerve has not been reported so far. NKB might be able to make the information interact accurately and effectively with NK-3R on the tumor through direct and indirect contact between cells or through the opening and closing of the signal transduction pathway, and it might also be able to initiate the recognition and response of the ligand and receptor in the perineural niche. However, there was no significant difference between the expression of NK-3R in cancer cells and the presence of lymphovascular invasion (*p* = 0.37) or perineural invasion (*p* = 0.44). It is suggested that overexpression of NK-3R in cancer cells in the invasion front leads to upregulation of the NK-3R signal, rather than the NKB ligand rising at the tumor invasion front due to increased lymphovascular and perineural invasion (Figure 5). The mechanism underlying the NKB signal in the tumor bone microenvironment remains to be examined in the future.

The involvement of the tachykinin receptor in tumorigenesis has been reported, and the receptor has been found to drug-block tachykinin receptor signaling in cancer cell lines of colon cancer, breast cancer, hepatoblastoma, leukemia, and neuroblastoma [46,47,48,49]. The biological actions of undecapeptide substance P (SP), originating from the *TAC1* gene, belong to the tachykinin family of peptides. These actions are mediated through binding to neurokinin receptors; specifically, members of the GPCR family, including NK-1R, NK-2R, and NK-3R, conferred the highest affinity binding for NK-1R [50]. However, this receptor family, especially members NK-2R and NK-3R, has not been fully studied for malignant tumors. Many types of tumor cells may express NKRs, including NK-3R, and tachykinin produced by tumor cells, microvessels, infiltrative nerves, or immune cells acts in an autocrine, paracrine, or neuroclinical manner, causing tumor cell proliferation and apoptosis.

The association between inflammation and NK-3R has been reported, and inflammation induced by the intraplantar injection of Formalin or an adjuvant upregulates NK-3R mRNA expression in the spinal dorsal horn [51]. Inflammation is a repair reaction that restores tissue damage from infection and other causes. Even in the tumor microenvironment, cancer and wound healing share several molecular pathways, and continuous cell regeneration and proliferation are induced by persistent inflammation [52]. In the tumor bone microenvironment, the RANK/OPG system is out of balance, which can lead to increased bone resorption and local inflammation [53]. Inflammation in or near bone induces a robust osteoclastogenic response, and the relative roles of RANKL and inflammatory cytokines such as TNFα, Il-1β, and Il-6 in this context have been debated [54]. Mice globally lacking RANKL with serum transfer arthritis showed comparable inflammation loss to controls; there was no osteoclast formation in affected joints, and bone erosion was minimal [55]. These findings indicated that RANKL plays a strong role in inflammation-induced bone resorption. An acute innate immune mediator, C-reactive protein (CRP), works with complement components to assist in the removal of foreign pathogens by binding with phosphocholine on the membrane of the pathogen. Elevated CRP is not just an early biomarker, but is likely a pathogenic factor contributing to preeclampsia by binding to phosphocholinated NKB and preferentially activating NK-3R [56]. Elevated CRP and NKB/NK-3R signaling contributes to the development of preeclampsia, and NKB/NK-3R signaling could be a therapeutic target and pathogenic biomarker for preeclampsia [57]. CRP levels showed a graded increase according to the severity of OSCC patients [58]. CRP level correlated with local and lymph node invasion [59], which could be regarded as a prognostic marker in OSCC patients [60,61]. CRP is a possible marker for the diagnosis and prognosis of OSCC [62], and it is suggested to be related to the NK-3R signal in the OSCC bone marrow environment, but the correlation between CRP and NK-3R expression was not seen in this study. Further studies are needed in order to elucidate the mechanisms underlying OSCC.

Due to the current lack of detailed knowledge about the effects of NK-3R on physiologycal bone metabolism [63,64,65], we are still unable to explain the tumor bone microenvironment. It will be a challenge to elucidate the relationship between the tumor bone microenvironment and neural regulation. Future research should explore areas related to how the tumor bone microenvironment regulates the stem cells; the interaction of NK-3R with other neuropeptides, cytokines, and hormones; and the potential role of NK-3R antagonists as effective therapeutic agents [39,63].

## Figures and Tables

**Figure 1 diagnostics-11-01044-f001:**
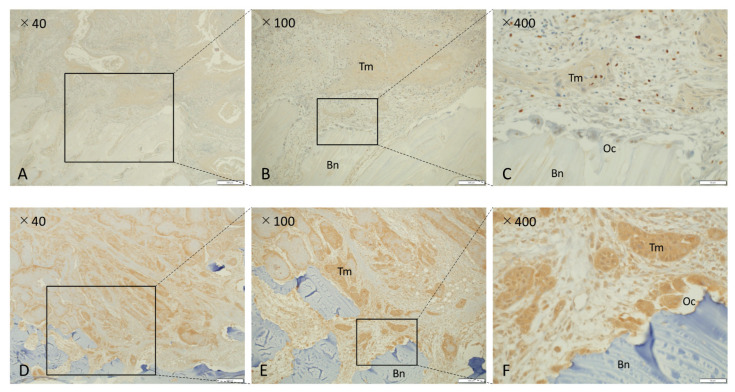
Immunohistochemical analysis of NK-3R in osteolytic mandibular SCCs. (**A**–**C**) NK-3R weak expression group. (**D**–**F**) NK-3R strong expression group. Each photo on the right is a magnification of the rectangle-delimited area of cancer bone invasion in the corresponding left photo. Scale bar: 500 μm (**A**,**D**), 200 μm (**B**,**E**), 50 μm (**C**,**F**). Bn: bone, Oc: osteoclast, Tm: tumor.

**Figure 2 diagnostics-11-01044-f002:**
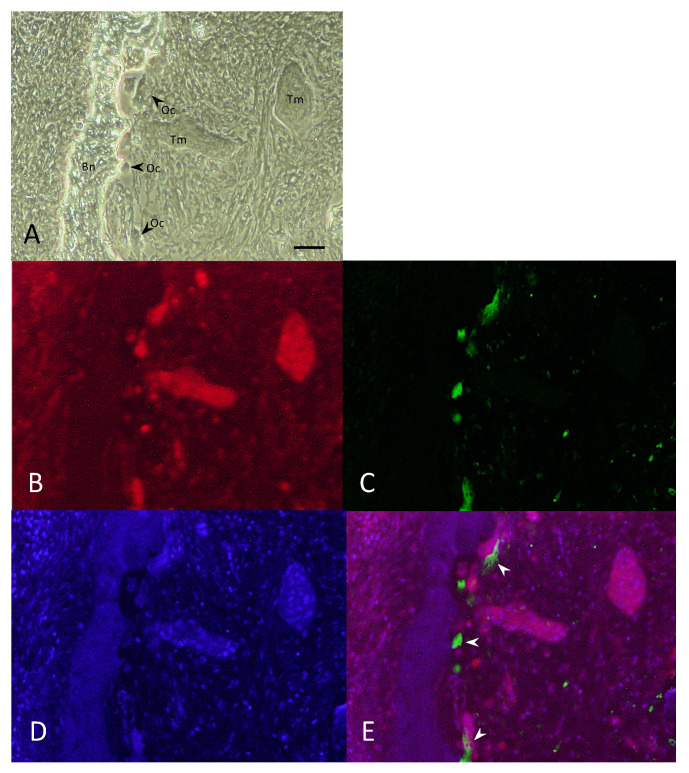
Double-fluorescent immunohistochemical analysis of NK-3R and CD68 in osteolytic mandibular SCCs. (**A**) Bright field image. (**B**–**E**) Double-fluorescent IHC for NK-3R (**B**), CD68 (**C**), DAPI (**D**), and Merge (**E**, arrowhead). Scale bar: 50 μm. Bn: bone, Oc: osteoclast, Tm: tumor.

**Figure 3 diagnostics-11-01044-f003:**
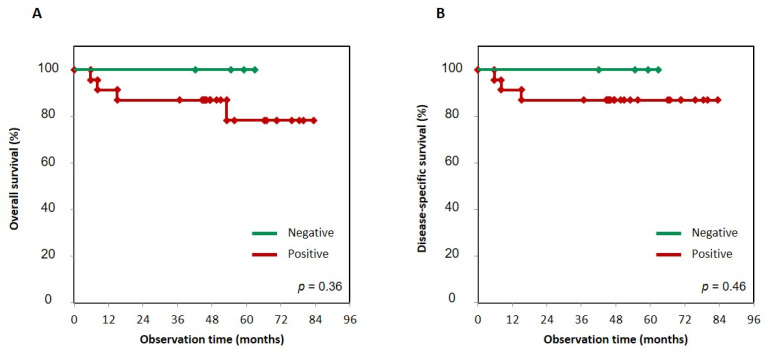
Kaplan–Meier curves of NK-3R-positive and -negative groups in gingival SCC patients. Overall survival (**A**) and disease-specific survival (**B**) were analyzed using the Kaplan–Meier method, and the difference in follow-up time between the two groups with positive and negative NK-3R expression was compared using the log-rank test. Patients with positive NK-3R tumors in overall survival and disease-specific survival had no correlation with patients who were negative for NK-3R tumors. *p* = 0.36 and 0.46, respectively.

**Figure 4 diagnostics-11-01044-f004:**
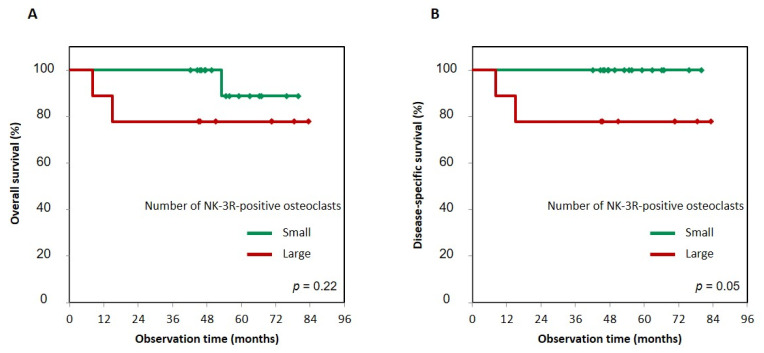
Kaplan–Meier curves of large and small groups of NK-3R-positive osteoclasts numbers in gingival SCC patients. Overall survival (**A**) and disease-specific survival (**B**) of large or small numbers of NK-3R-positive osteoclasts. Patients with a large number of positive NK-3R osteoclasts tended to show lower overall survival than patients with a small number, *p* = 0.22 (**A**). On the other hand, patients with a large number of positive NK-3R osteoclasts showed a significantly greater decrease in disease-specific survival than patients with a small number, *p* = 0.05 (**B**). Photographs of three visual fields in the cancer bone destruction area were taken at a magnification of ×400. The number of osteoclasts per visual field was averaged; “large” was defined as three or more NK-3R-positive cells, while “small” was defined as two or fewer positive cells.

**Figure 5 diagnostics-11-01044-f005:**
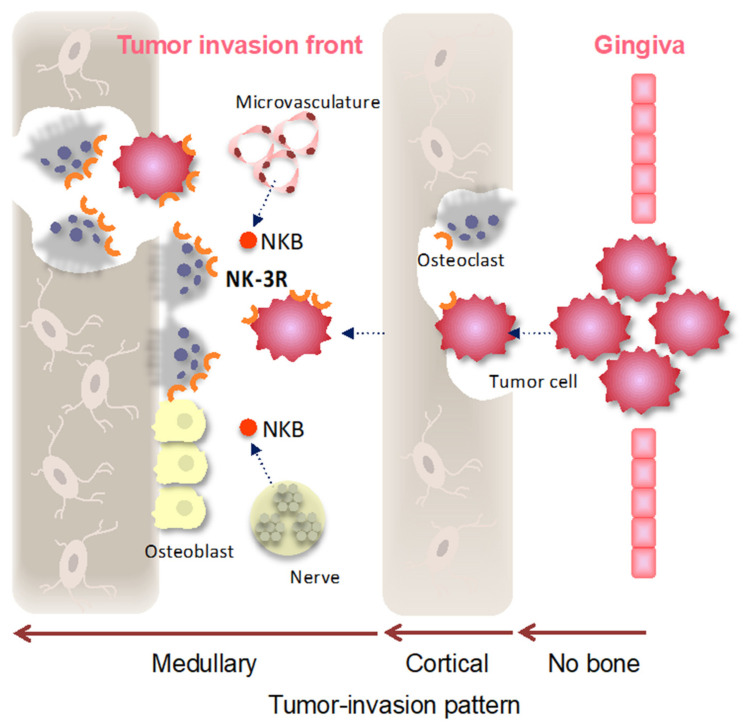
Cartoon representing NK-3R signaling in gingival SCC-induced bone destruction. Gingival SCC arises from the mucosal surface of the oral gingiva, clinically showing three types of mandibular destruction patterns. The type that does not destroy cortical bone (no bone), the type that shows cortical bone invasion (the bone invasion was limited to the cortex; cortical), the type that shows medullary bone invasion (medullary), and the type that shows both cortical and medullary bone invasion are involved in the presence of osteoclasts. After the gingival SCC destroys the cortical bone and infiltrates the medullary bone, NK-3R expression in the tumor invasion front and osteoclasts might be induced by various growth factors and by chemokine-including NKB ligand produced from the microvasculature and nerves in the special environment of the medullary.

**Table 1 diagnostics-11-01044-t001:** Clinicopathological characteristics of gingival SCC patients.

Parameter		Number (N = 27)	%
Age (years)	39–89 (mean 72.8)		
Gender	Male	10	37.0
	Female	17	63.0
Tumor site	Upper gingiva	10	37.0
	Lower gingiva	17	63.0
Tumor size	<2 cm	4	14.8
	≥2 cm, <4 cm	17	63.0
	≥4 cm	6	22.2
Pathological N stage	N0	22	81.5
	N1	3	11.1
	N2	2	7.4
Pathological TNM stage	I	3	11.1
	II	13	48.1
	III	1	3.7
	IVA	9	33.3
	IVB	1	3.7
Differentiation	Well	15	55.6
	Moderate	9	33.3
	Poor	3	11.1

**Table 2 diagnostics-11-01044-t002:** Relationship between NK-3R protein expression in tumor cells and clinicopathological characteristics in gingival SCC.

Parameter		NK-3R Expression in Tumor Cells		*p*-Value
		Positive (N = 23)	Negative (N = 4)	
Age	<70	7	2	0.41
	≥70	16	2	
Gender	Male	7	3	0.13
	Female	16	1	
Tumor size	<2 cm	2	2	0.08
	≥2 cm, <4 cm	16	1	
	≥4 cm	5	1	
N stage	N0	18	4	0.59
	N1	3	0	
	N2	2	0	
TNM Stage	I, II	13	3	0.49
	III, IV	10	1	
Differentiation	Well	11	4	0.08
	Moderate–poor	12	0	
Bone invasion	No bone invasion	7	4	0.03
	Cortical	8	0	
	Medullary	8	0	
Lymphovascular invasion	Absent	19	4	0.37
	Present	4	0	
Perineural invasion	Absent	20	4	0.44
	Present	3	0	
Primary recurrence	Absent	17	3	0.96
	Present	6	1	
Secondary cervical lymph nodal metastasis	Absent	20	4	0.61
	Present	3	0	

**Table 3 diagnostics-11-01044-t003:** Relationship between the number of NK-3R positive osteoclasts and clinicopathological characteristics.

Parameter		‘NK-3R-Positive’ Osteoclast Number		*p*-Value
		Large (N = 9)	Small (N = 16)	
Age	<70	2	7	0.28
	≥70	7	9	
Gender	Male	1	7	0.09
	Female	8	9	
Tumor size	<2 cm	2	2	0.49
	≥2 cm, <4 cm	4	11	
	≥4 cm	3	3	
N stage	N0	6	15	0.12
	N1	1	1	
	N2	2	0	
TNM Stage	I, II	3	13	0.016
	III, IV	6	3	
Differentiation	Well	5	9	0.97
	Moderate–poor	4	7	
Bone invasion	No bone invasion	0	11	<0.001
	Cortical	3	5	
	Medullary	6	0	
Lymphovascular invasion	Absent	6	15	0.07
	Present	3	1	
Perineural invasion	Absent	7	15	0.23
	Present	2	1	
Primary recurrence	Absent	6	12	0.71
	Present	3	4	
Secondary cervical lymph nodal metastasis	Absent	8	14	0.91
	Present	1	2	
NK-3R expression in tumor cells	Positive	9	12	0.1
	Negative	0	4	

## Data Availability

All supporting data are provided in the current manuscript.

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
