# Peer review of "Expression of Neurokinin B Receptor in the Gingival Squamous Cell Carcinoma Bone Microenvironment"

_diagnostics, 2021, doi:10.3390/diagnostics11061044_

Round 1
Reviewer 1 Report
Expression of Neurokinin B Receptor Induced by Hedgehog Signaling in the Gingival Squamous Cell Carcinoma Bone Microenvironment
Summary
The main question addressed by the research is: what is the relationship between NK-3R expression and the prognosis of gingival SCC patients? Previously the authors found that expression of NK-3R is detected in osteoclasts and tumor cells at the invasion front, and thus posed that NK-3R signaling in the gingival SCC bone microenvironment plays an important role in tumor. Through IHC analysis of a total of 27 gingival cancer patients, the research concludes that (1) higher NK-3R expression in tumor cells was found in group with jaw bone invasion, but no association was noted between NK-3R and tumor size, stage, or differentiation. (2) A trend of poor survival in NK-3R positive cases (without significance). (3) larger number of NK-3R-positive osteoclasts was associated with stage, bone invasion, and DFS.
Major Issues
Some problems are addressed as following.
- The title: Please ensure that the paper title is accurate and truly representative of the content of the paper. The present study only showed the analysis of NK-3R in cancer patients, it’s not proper to say that neurokinin B receptor induced by hedgehog signaling here since there was no proof in this study. Correlatively, in Line 158-159, the authors stated that “Our previous studies together suggested that NK-3R expression in osteoclasts is activated by tumor express SHH in the tumor bone microenvironment”, but there was no proper citation. I went through the shown references one by one but still could not find the link between SHH and NK-3R in tumors.
- Detail of methodology:
- (line 116-117) How about <50% of tumor cells showed strong staining and >=50% cells showed weak staining? If staining percentage and intensity are quantified together, the presentation should be a labeling score, or here at least 4 groups should be shown (<50% weak, >50% weak, <50% strong, > 50% strong)
- (line 118-119) How to define osteoclasts? Only by HE examination? Double-staining with markers of osteoclasts should be more accurate and specific.
- (line 118-124) Would the counting of osteoclasts be interfered with tissue embedding and section? If we don’t see any osteoclasts in the field, that doesn’t mean there were no cells surrounding the invasion front. To use the percentage of NK-3R-positive osteoclasts (positive osteoclasts/total osteoclasts) here should be better than absolute cell numbers. Why did the authors choose > 3 osteoclasts for a large number, instead of 4 or 5?
- Analytical issue: please indicate the number of collected specimens and the rationale for determining such number. A sample number of n=27 does not seem adequate to perform statistical evaluation in survival analysis.
- The explanation: TNM staging of oral cancer defines that tumor invading through cortical bone should be a T4a case. Thus, the classification of TNM stage and bone invasion in Table 2 of the present study should be nearly the same since there was only 1 case of Stage III. The number of Stage I+II and III+IV was 16 and 11, but the number of no bone invasion and bone invasion was 11 and 16. Why does the discrepancy exist? (That is, the 16 cases with bone invasion should be defined as Stage IV, but here only 10 cases are defined as Stage IV in Table 1 and 2).
- The interpretation: same with point 1, it is not proper to pose the model with SHH in Figure 4. If so, please provide the data with SHH in the results.
- The citation: the authors cited quite a few their own publications. However, some of them are only descriptive study showing a staining phenomenon, and the sample size is even too small (such as n=9) to carry out a statistical analysis. Using such reference to pose a hypothesis or conclusion is not convincing.
Minor Issues
- Typing errors: (line 263, 276, 296) The medial bone invasion should be the medullary bone invasion.
Author Response
REVIEWER #1:
We greatly appreciate the reviewer’s insightful comments regarding our manuscript and are thankful for your considering our paper acceptable in accordance with the reviewers' comments. We have answered all of the reviewers’ comments below, added the text, performed immunohistochemical analysis, added the newly Figure 2 and changed the illustration in Figure 4 (newly Figure 5) accordingly and hope that you will now find the manuscript acceptable for publication in Diagnostics.
Comment #1: The title: Please ensure that the paper title is accurate and truly representative of the content of the paper. The present study only showed the analysis of NK-3R in cancer patients, it’s not proper to say that neurokinin B receptor induced by hedgehog signaling here since there was no proof in this study. Correlatively, in Line 158-159, the authors stated that “Our previous studies together suggested that NK-3R expression in osteoclasts is activated by tumor express SHH in the tumor bone microenvironment”, but there was no proper citation. I went through the shown references one by one but still could not find the link between SHH and NK-3R in tumors.
Response #1: We have changed the tile “Expression of Neurokinin B Receptor in the Gingival Squamous Cell Carcinoma Bone Microenvironment” according to the reviewer’s comments.
We have deleted the text.“Our previous studies together suggested that NK-3R expression in osteoclasts is activated by tumor express SHH in the tumor bone microenvironment”
Please allow us to include related three references in other sections, “9. PLoS One 2016, 11, (3), e0151731. Please refer to Supplemental Figure2”, “15. Anticancer Res. 2016, 36, (12), 6335-6341. Tachykinin Receptor 3 (NK-3R) expression in OSCC invasion front.” and “16. Anticancer Res. 2017, 37, (11), 6119-6123. NK-3R signaling is a potential target for the treatment of oral squamous cell carcinoma in cases of bone destruction” in the Results section.
Comment #2: Detail of methodology: (line 116-117) How about <50% of tumor cells showed strong staining and >=50% cells showed weak staining? If staining percentage and intensity are quantified together, the presentation should be a labeling score, or here at least 4 groups should be shown (<50% weak, >50% weak, <50% strong, > 50% strong)
Response #2: Thank you for your comments. It should be originally shown in 4 groups (<50% weak, >50% weak, <50% strong, > 50% strong) as you suggested. In this analysis, there were no cases of ≥50% weak and <50% strong, so we classified as “negative, <50% of tumor cells showed weak to moderate immunoreactivity; positive, ≥50% of tumor cells showed strong immunoreactivity”. We have changed the text as “<50% weak, ≥50% weak and <50% strong were classified as negative, and> 50% strong was classified as positive” in the section of “Quantification of NK-3R in tissue sections in Materials and Methods”.
Comment #3: Detail of methodology: (line 118-119) How to define osteoclasts? Only by HE examination? Double-staining with markers of osteoclasts should be more accurate and specific.
Response #3: Thank you for your comments. We have performed the double-staining with markers of osteoclasts CD68 in the resected clinical samples, confirmed the number of osteoclasts were accurate. We have added the information to define osteoclasts in the “Quantification of NK-3R in tissue sections”, newly “Double-fluorescent immunohistochemical analysis” in Materials and Methods, Results and newly Figure 2.
Quantification of NK-3R in tissue sections.
"The NK-3R-positive osteoclasts were counted by taking a x400 randomly photographed tumor bone invasion front as one field of view, counting the NK-3R and CD68 double-positive osteoclasts contained in it, and averaging the three fields of view."
Double-fluorescent immunohistochemical analysis
Double-fluorescent IHC for NK-3R-CD68 was performed using anti-NK-3R antibody (1:100) (#bs-0166R, anti-rabbit IgG, Bioss) and anti-CD68 (1:100) (#KP1, anti-mouse IgG, Dako, Glostrup, Denmark). The secondary antibodies applied are anti-rabbit IgG Alexa Flour 568 and anti-mouse IgG Alexa Flour 488 (Thermo Fisher, Tokyo, Japan). Antibodies were diluted with Can Get Signal A (TOYOBO, Osaka, Japan). After antigen retrieval, sections were treated with Block Ace (DS Pharma Bio-medical, Osaka, Japan) for 20 min at room temperature. Specimens were incubated with primary antibodies at 4C overnight and then incubated with secondary antibodies (1:200) for 1 h at room temperature. After the reaction, the specimens were stained with 1 mg/mL DAPI (Dojindo Laboratories, Kumamoto, Japan).
Results
“To determine whether the NK-3R is accurate and specific express in the osteoclasts at the invasive bone destruction site observed in patients with oral squamous cell carcinoma, we performed double-fluorescent IHC for NK-3R-CD68 (Fig. 2A-D). As shown in Figure 2, the expression of NK-3R was observed not only in the tumor cells (Fig. 2A Tm and Fig. 2B) but also in the almost osteoclasts (Fig. 2A OC and Fig. 2B) which were correlated with the expression of CD68 positive cells (Fig. 2C and E, arrow heads) at the tumor bone invasion front.”
Comment #4: Detail of methodology: (line 118-124) Would the counting of osteoclasts be interfered with tissue embedding and section? If we don’t see any osteoclasts in the field, that doesn’t mean there were no cells surrounding the invasion front. To use the percentage of NK-3R-positive osteoclasts (positive osteoclasts/total osteoclasts) here should be better than absolute cell numbers. Why did the authors choose > 3 osteoclasts for a large number, instead of 4 or 5?
Response #4: Thank you for your comments. To use the percentage of NK-3R-positive osteoclasts (positive osteoclasts/total osteoclasts) than absolute cell numbers. We have found that the expression of NK-3R was observed in the almost osteoclasts (Fig. 2A OC and Fig. 2B) which were correlated with the expression of CD68 positive cells (Fig. 2C and E) at the tumor bone invasion front. For this reason, we chose the cell number and the description was added in the text.
We have chosen the presence of 3 or more NK-3R-positive osteoclasts in one visual field was defined as a large number of cases under the x400 magnification. We have added the text according to the comments in the section of “Quantification of NK-3R in tissue sections in Materials and Methods” and the Results.
“The NK-3R-positive osteoclasts were counted by taking a x400 randomly photographed tumor bone invasion front as one field of view, counting the NK-3R and CD68 double-positive osteoclasts”
“All the multiple serial sections in a clinical specimen were checked by surgeons and pathologists, confirming that the double-positive analyzed data were consistent with HE data. Figure 1 and 2 shows the results of representative NK-3R and CD68 staining in tumor bone invasion front.”
“To determine whether the NK-3R is accurate and specific express in the osteoclasts at the invasive bone destruction site observed in patients with oral squamous cell carcinoma, we performed double-fluorescent IHC for NK-3R-CD68 (Fig. 2A-D). As shown in Figure 2, the expression of NK-3R was observed not only in the tumor cells (Fig. 2A Tm and Fig. 2B) but also in the almost osteoclasts (Fig. 2A OC and Fig. 2B) which were correlated with the expression of CD68 positive cells (Fig. 2C and E) at the tumor bone invasion front.”
Comment #5: Analytical issue: please indicate the number of collected specimens and the rationale for determining such number. A sample number of n=27 does not seem adequate to perform statistical evaluation in survival analysis.
Response #5: Thank you for your comments. We have added the text in the Discussion section.
“However, regarding the examination of the relationship between the NK-3R expression and prognosis in OSCC patients, it is necessary to further increase the number of cases and examine it in detail in the future.”
Comment #6: The explanation: TNM staging of oral cancer defines that tumor invading through cortical bone should be a T4a case. Thus, the classification of TNM stage and bone invasion in Table 2 of the present study should be nearly the same since there was only 1 case of Stage III. The number of Stage I+II and III+IV was 16 and 11, but the number of no bone invasion and bone invasion was 11 and 16. Why does the discrepancy exist? (That is, the 16 cases with bone invasion should be defined as Stage IV, but here only 10 cases are defined as Stage IV in Table 1 and 2).
Response #6: Thank you for your comments. UICC, 8th TNM classification describe that T4a: Tumor invades through the cortical bone of the mandible or maxillary sinus, or invades the skin of the face. In this study, cortical was described as when the bone invasion was limited to the cortex, and the following text was added at the Histopathological analysis in the section of Patients and Methods and figure legend, and the illustration of the cortical was changed in Figure 4 (newly Figure 5).
“In this study the cortical is not classified as T4a because tumor does not invade through the cortical bone.”
“the type that shows cortical bone invasion (the bone invasion was limited to the cortex) (Cortical)”
Comment #7: The interpretation: same with point 1, it is not proper to pose the model with SHH in Figure 4. If so, please provide the data with SHH in the results.
Response #7: Thank you for your comments. SHH was deleted from Figure 4 (newly Figure 5) according to the comments.
Comment #8: The citation: the authors cited quite a few their own publications. However, some of them are only descriptive study showing a staining phenomenon, and the sample size is even too small (such as n=9) to carry out a statistical analysis. Using such reference to pose a hypothesis or conclusion is not convincing.
Response #9: Thank you for your comments. We have changed the tile “Expression of Neurokinin B Receptor in the Gingival Squamous Cell Carcinoma Bone Microenvironment” and SHH was deleted from Figure 4 (newly Figure 5) according to the reviewer’s comments.
We have deleted the text.“Our previous studies together suggested that NK-3R expression in osteoclasts is activated by tumor express SHH in the tumor bone microenvironment”
Please allow us to include related three references in other sections, “9. PLoS One 2016, 11, (3), e0151731. Please refer to Supplemental Figure2”, “15. Anticancer Res. 2016, 36, (12), 6335-6341. Tachykinin Receptor 3 (NK-3R) expression in OSCC invasion front.” and “16. Anticancer Res. 2017, 37, (11), 6119-6123. NK-3R signaling is a potential target for the treatment of oral squamous cell carcinoma in cases of bone destruction” in the Results section.
We also added the new conclusion and other references at the end of Discussion.
New references
- Li, F.X.; Xu, F.; Lin, X.; Wu, F.; Zhong, J.Y.; Wang, Y.; Guo, B.; Zheng, M.H.; Shan, S.K.; Yuan, L.Q., The Role of Substance P in the Regulation of Bone and Cartilage Metabolic Activity. Front Endocrinol (Lausanne). 2020, 11:77.
- Tatullo, M.; Spagnuolo, G.; Codispoti, B.; Zamparini, F.; Zhang, A.; Degli Esposti, M.; Aparicio, C.; Rengo, C.; Nuzzolese, M.; Manzoli, L.; Fava, F.; Prati, C.; Fabbri, P.; Gandolfi, M.G., PLA-Based Mineral-Doped Scaffolds Seeded with Human Periapical Cyst-Derived MSCs: A Promising Tool for Regenerative Healing in Dentistry. Materials (Basel) 2019,12, (4), 597.
- Marrelli, M.; Codispoti, B.; Shelton, R.M.; Scheven, B.A.; Cooper, P.R.; Tatullo, M.; Paduano, F.; Dental Pulp Stem Cell Mechanoresponsiveness: Effects of Mechanical Stimuli on Dental Pulp Stem Cell Behavior. Front Physiol. 2018, 9, 1685.
- Wang, X.D.; Li, S.Y.; Zhang, S.J.; Gupta, A.; Zhang, C.P.; Wang, L., The neural system regulates bone homeostasis via mesenchymal stem cells: a translational approach. Theranostics. 2020, 10, (11), 4839-4850.
Minor Issues
Comment #1: Typing errors: (line 263, 276, 296) The medial bone invasion should be the medullary bone invasion.
Response #1 We have changed the words.
Reviewer 2 Report
The paper entitled “Expression of Neurokinin B Receptor Induced by Hedgehog Signaling in the Gingival Squamous Cell Carcinoma Bone Microenvironment” is interesting.
The authors refer about “Bone microenvironment” but no studies on this microenvironment in physiological processes have been reported, to compare the golden conditions in clinical situation. Moreover, and more importantly, they must discuss on impact of biomaterials/scaffolds on bone cell growth (Please, see “Tatullo M, Spagnuolo G, Codispoti B, Zamparini F, Zhang A, Esposti MD, Aparicio C, Rengo C, Nuzzolese M, Manzoli L, Fava F, Prati C, Fabbri P, Gandolfi MG. PLA-Based Mineral-Doped Scaffolds Seeded with Human Periapical Cyst-Derived MSCs: A Promising Tool for Regenerative Healing in Dentistry. Materials (Basel). 2019 Feb 16;12(4):597.”) - AND - (Please, see and also discuss “Marrelli M, Codispoti B, Shelton RM, Scheven BA, Cooper PR, Tatullo M, Paduano F. Dental Pulp Stem Cell Mechanoresponsiveness: Effects of Mechanical Stimuli on Dental Pulp Stem Cell Behavior. Front Physiol. 2018 Nov 26;9:1685.”)
Poor has been reported on the role of local stem cells, which can biologically act as the immunomodulatory and pro-regenerative activities in the local environment (Please, see and discuss: Ballini A, Boccaccio A, Saini R, Van Pham P, Tatullo M. Dental-Derived Stem Cells and Their Secretome and Interactions with Bioscaffolds/Biomaterials in Regenerative Medicine: From the In Vitro Research to Translational Applications. Stem Cells Int. 2017;2017:6975251.).
Minor suggestion:
Author have reported that “NK-3R, a tachykinin receptor, is a GPCR that is preferentially activated by hedgehog agonists in osteoclasts [9]. In addition to being involved in a wide range of CNSs, it can exert many biological effects including smooth muscle contraction and relaxation, vaso-dilation, immune system activation, pain transmission, and neurogenic inflammation [19].”: In this context, authors may also briefly improve the discussion on the role of natural compounds and their impact on local inflammation in both clinical and oncological conditions (Please, see and discuss: “Inchingolo F, Tatullo M, Marrelli M, Inchingolo AM, Picciariello V, Inchingolo AD, Dipalma G, Vermesan D, Cagiano R. Clinical trial with bromelain in third molar exodontia. Eur Rev Med Pharmacol Sci. 2010 Sep;14(9):771-4. PMID: 21061836.”)
Finally, Conclusions should be improved with clear take-home messages.
Author Response
REVIEWER #2:
We greatly appreciate the reviewer’s insightful comments regarding our manuscript and are thankful for your considering our paper acceptable in accordance with the reviewers' comments. We have answered all of the reviewers’ comments below and modified the text and added references accordingly and hope that you will now find the manuscript acceptable for publication in Diagnostics.
Results and discussion:
Comment #1: The authors refer about “Bone microenvironment” but no studies on this microenvironment in physiological processes have been reported, to compare the golden conditions in clinical situation. Moreover, and more importantly, they must discuss on impact of biomaterials/scaffolds on bone cell growth (Please, see “Tatullo M, Spagnuolo G, Codispoti B, Zamparini F, Zhang A, Esposti MD, Aparicio C, Rengo C, Nuzzolese M, Manzoli L, Fava F, Prati C, Fabbri P, Gandolfi MG. PLA-Based Mineral-Doped Scaffolds Seeded with Human Periapical Cyst-Derived MSCs: A Promising Tool for Regenerative Healing in Dentistry. Materials (Basel). 2019 Feb 16;12(4):597.”) - AND - (Please, see and also discuss “Marrelli M, Codispoti B, Shelton RM, Scheven BA, Cooper PR, Tatullo M, Paduano F. Dental Pulp Stem Cell Mechanoresponsiveness: Effects of Mechanical Stimuli on Dental Pulp Stem Cell Behavior. Front Physiol. 2018 Nov 26;9:1685.”)
Response #1: Thank you for your comments. The references were added in the Discussion section according to the comments.
- Tatullo, M.; Spagnuolo, G.; Codispoti, B.; Zamparini, F.; Zhang, A.; Degli Esposti, M.; Aparicio, C.; Rengo, C.; Nuzzolese, M.; Manzoli, L.; Fava, F.; Prati, C.; Fabbri, P.; Gandolfi, M.G., PLA-Based Mineral-Doped Scaffolds Seeded with Human Periapical Cyst-Derived MSCs: A Promising Tool for Regenerative Healing in Dentistry. Materials (Basel) 2019,12, (4), 597.
- Marrelli, M.; Codispoti, B.; Shelton, R.M.; Scheven, B.A.; Cooper, P.R.; Tatullo, M.; Paduano, F.; Dental Pulp Stem Cell Mechanoresponsiveness: Effects of Mechanical Stimuli on Dental Pulp Stem Cell Behavior. Front Physiol. 2018, 9, 1685.
Comment #2: Poor has been reported on the role of local stem cells, which can biologically act as the immunomodulatory and pro-regenerative activities in the local environment (Please, see and discuss: Ballini A, Boccaccio A, Saini R, Van Pham P, Tatullo M. Dental-Derived Stem Cells and Their Secretome and Interactions with Bioscaffolds/Biomaterials in Regenerative Medicine: From the In Vitro Research to Translational Applications. Stem Cells Int. 2017;2017:6975251.).
Response #2: Thank you for your comments. I have read the manuscript you pointed out and tried to include it in the Reference, but we could not find any relevance to this research.
The following treatises have been newly added.
- Wang, X.D.; Li, S.Y.; Zhang, S.J.; Gupta, A.; Zhang, C.P.; Wang, L., The neural system regulates bone homeostasis via mesenchymal stem cells: a translational approach. Theranostics. 2020, 10, (11), 4839-4850.
Comment #3: Author have reported that “NK-3R, a tachykinin receptor, is a GPCR that is preferentially activated by hedgehog agonists in osteoclasts [9]. In addition to being involved in a wide range of CNSs, it can exert many biological effects including smooth muscle contraction and relaxation, vaso-dilation, immune system activation, pain transmission, and neurogenic inflammation [19].”: In this context, authors may also briefly improve the discussion on the role of natural compounds and their impact on local inflammation in both clinical and oncological conditions (Please, see and discuss: “Inchingolo F, Tatullo M, Marrelli M, Inchingolo AM, Picciariello V, Inchingolo AD, Dipalma G, Vermesan D, Cagiano R. Clinical trial with bromelain in third molar exodontia. Eur Rev Med Pharmacol Sci. 2010 Sep;14(9):771-4. PMID: 21061836.”)
Response #3: Thank you for your comments. I have read the manuscript you pointed out and tried to include it in the Reference, but we could not find any relevance to this research.
The following treatises have been newly added.
- Li, F.X.; Xu, F.; Lin, X.; Wu, F.; Zhong, J.Y.; Wang, Y.; Guo, B.; Zheng, M.H.; Shan, S.K.; Yuan, L.Q., The Role of Substance P in the Regulation of Bone and Cartilage Metabolic Activity. Front Endocrinol (Lausanne). 2020, 11:77.
Comment #4: Conclusions should be improved with clear take-home messages.
Response #4: Thank you for your comments. We also added the new conclusion and new references at the end of Discussion.
- Li, F.X.; Xu, F.; Lin, X.; Wu, F.; Zhong, J.Y.; Wang, Y.; Guo, B.; Zheng, M.H.; Shan, S.K.; Yuan, L.Q., The Role of Substance P in the Regulation of Bone and Cartilage Metabolic Activity. Front Endocrinol (Lausanne). 2020, 11:77.
- Tatullo, M.; Spagnuolo, G.; Codispoti, B.; Zamparini, F.; Zhang, A.; Degli Esposti, M.; Aparicio, C.; Rengo, C.; Nuzzolese, M.; Manzoli, L.; Fava, F.; Prati, C.; Fabbri, P.; Gandolfi, M.G., PLA-Based Mineral-Doped Scaffolds Seeded with Human Periapical Cyst-Derived MSCs: A Promising Tool for Regenerative Healing in Dentistry. Materials (Basel) 2019,12, (4), 597.
- Marrelli, M.; Codispoti, B.; Shelton, R.M.; Scheven, B.A.; Cooper, P.R.; Tatullo, M.; Paduano, F.; Dental Pulp Stem Cell Mechanoresponsiveness: Effects of Mechanical Stimuli on Dental Pulp Stem Cell Behavior. Front Physiol. 2018, 9, 1685.
- Wang, X.D.; Li, S.Y.; Zhang, S.J.; Gupta, A.; Zhang, C.P.; Wang, L., The neural system regulates bone homeostasis via mesenchymal stem cells: a translational approach. Theranostics. 2020, 10, (11), 4839-4850.
Reviewer 3 Report
Dear Editor, I think that this is an interesting topics, well described by the authors.
Author Response
Thank you very much for your evaluation.
Round 2
Reviewer 1 Report
Comments
- The title: why is “receptor” deleted in title? The present study surely focused on NK-3R, but not NKB. Please recheck your title, and correct it if it’s a typing error.
- The authors have incorporated most of the comments in the revised version, including definition of positive staining, double-staining for osteoclasts, and explanation of classification of bone invasion.
- The proposed model: in Figure 5, the author stated that “It is suggested that NK-3R expression in the tumor invasion front and osteoclasts are upregulated by the SHH ligand”, and “SHH ligand produced by tumor cells induces production of the receptor activator of nuclear factor kB ligand (RANKL) in osteoblasts and indirectly promotes osteoclast formation”. My questions: (1) if tumor-derived SHH ligand could indirectly activated osteoclast and thus promote cell invasion, what is the role of NK-3R here? The meaning of upregulation of NK-3R in tumor and osteoclast was not discussed/posed in this figure legend. (2) It seems that all events happened in medullary bone in this model. Why didn’t the phenomenon exist in cortical bone since tumor-derived SHH ligand were still present in the microenvironment? No cortical erosion, no medullary invasion. The tumor cells should have equivalent aggressiveness in both cortical and medullary bone which driven the bone invasion. (3) Any chemoattractant effects of NK-3R in this model? I think this figure is too rough and not convincing because of absence of in intro
Author Response
We greatly appreciate the reviewer’s insightful comments regarding our manuscript and are thankful for your considering our paper acceptable in accordance with the reviewers' comments. We have answered all of the reviewers’ comments below and added the text accordingly and hope that you will now find the manuscript acceptable for publication in Diagnostics.
Comment #1: The title: why is “receptor” deleted in title? The present study surely focused on NK-3R, but not NKB. Please recheck your title, and correct it if it’s a typing error.
Response #1: Thank you for your comments. It was a typing error. We have changed the tile “Expression of Neurokinin B Receptor in the Gingival Squamous Cell Carcinoma Bone Microenvironment” according to the reviewer’s comments.
Comment #2: The authors have incorporated most of the comments in the revised version, including definition of positive staining, double-staining for osteoclasts, and explanation of classification of bone invasion.
Response #2: Thank you for your comments. The quality of the manuscript was improved by your comments.
Comment #3: The proposed model: in Figure 5, the author stated that “It is suggested that NK-3R expression in the tumor invasion front and osteoclasts are upregulated by the SHH ligand”, and “SHH ligand produced by tumor cells induces production of the receptor activator of nuclear factor kB ligand (RANKL) in osteoblasts and indirectly promotes osteoclast formation”. (1) if tumor-derived SHH ligand could indirectly activated osteoclast and thus promote cell invasion, what is the role of NK-3R here? The meaning of upregulation of NK-3R in tumor and osteoclast was not discussed/posed in this figure legend. (2) It seems that all events happened in medullary bone in this model. Why didn’t the phenomenon exist in cortical bone since tumor-derived SHH ligand were still present in the microenvironment? No cortical erosion, no medullary invasion. The tumor cells should have equivalent aggressiveness in both cortical and medullary bone which driven the bone invasion. (3) Any chemoattractant effects of NK-3R in this model? I think this figure is too rough and not convincing because of absence of in intro
Response #3: Thank you for your comments. We have changed the text according to your comments. (1) We have deleted the text about SHH in the Figure legends of Figure 5. (2) We have newly added the text that NK-3R is expressed in the special environment of medullary in the Figure legends of Figure 5. (3) The text was added in the Discussion section.
We have added (Figure 5) in the Discussion section.
It is suggested that overexpression of NK-3R in cancer cells in the invasion front leads to upregulation of the NK-3R signal rather than the NKB ligand rising at the tumor invasion front due to increased lymphovascular and perineural invasion (Figure 5).
We have changed the text in the Figure legends of Figure 5.
NK-3R expression in the tumor invasion front and osteoclasts is might be induced by various growth factors and by chemokine including NKB ligand produced from the microvasculature and nerves in the special environment of medullary. It is suggested that NK-3R expression in the tumor invasion front and osteoclasts are upregulated by the SHH ligand in autocrine and paracrine manners. The SHH ligand produced by tumor cells induces production of the receptor activator of nuclear factor kB ligand (RANKL) in osteoblasts and indirectly promotes osteoclast formation, or leads to increased osteoclast formation directly in the presence of RANKL. The tumor at the invasion front and osteoclasts activates NKB signaling by the NKB ligand produced from the microvasculature and nerves in the medullary.
Discussion section
It has also been suggested that the roles of NK-3R in tumor cell invasion and proliferation ability as well as osteoclast bone resorption act by different spatiotemporal mechanisms, because no significant correlation was found between the expression of NK-3R in cancer cells and the number of NK-3R-positive osteoclasts (Table3 p=0.01). The functional mechanism underlying the NKB signal in the tumor remains to be examined in the future.
Reviewer 2 Report
comments have been addressed
Author Response
We greatly appreciate the reviewer’s insightful comments regarding our manuscript and are thankful for your considering our paper acceptable in accordance with the reviewers' comments. We have answered all of the reviewers’ comments, modified the text and checked English accordingly and hope that you will now find the manuscript acceptable for publication in Diagnostics.
Round 3
Reviewer 1 Report
Comments
- The authors have incorporated most of the comments in this newly revised version, especially the Figure legend of Figure 5.
- If you could provide in vitro data in your future publication, it would strengthen the proposed model.
- The last question: you have counted the absolute number of NK-3R positive osteoclasts for calculation of survival analysis, and mentioned in the text that “NK-3R was observed also in the almost osteoclasts” (Line 213-214). So, can we just count the number of osteoclasts in H&E staining to predict patient prognosis, regardless of the expression of NK-3R?
Author Response
We greatly appreciate the reviewer’s insightful comments regarding our manuscript and are thankful for your considering our paper acceptable in accordance with the reviewers' comments. We have answered all of the reviewers’ comments below and added the text accordingly and hope that you will now find the manuscript acceptable for publication in Diagnostics.
Comment #1: The authors have incorporated most of the comments in this newly revised version, especially the Figure legend of Figure 5.
Response #1: Thank you for your comments. The quality of the manuscript was improved by your comments.
Comment #2: If you could provide in vitro data in your future publication, it would strengthen the proposed model.
Response #2: Thank you for your insightful comments. We would like to study in vitro data and connect it to future publications.
Comment #3: 3. The last question: you have counted the absolute number of NK-3R positive osteoclasts for calculation of survival analysis, and mentioned in the text that “NK-3R was observed also in the almost osteoclasts” (Line 213-214). So, can we just count the number of osteoclasts in H&E staining to predict patient prognosis, regardless of the expression of NK-3R?
Response #3: Thank you for your comments. We have added the text according to your comments in the Discussion section.
As shown in Figure 2, the expression of NK-3R was observed not only in the tumor cells but also in the almost osteoclasts which were correlated with the expression of CD68 positive cells at the tumor bone invasion front. However, regarding whether the prognosis of patients can be predicted simply by counting the number of osteoclasts by H & E staining regardless of the expression of NK-3R, further detailed study is needed because these data were included both weak and strong expression of NK-3R in osteoclasts.